# Treatment of Metastatic Disease through Natural Killer Cell Modulation by Infected Cell Vaccines

**DOI:** 10.3390/v11050434

**Published:** 2019-05-11

**Authors:** Seyedeh Raheleh Niavarani, Christine Lawson, Lee-Hwa Tai

**Affiliations:** 1Department of Anatomy and Cell Biology, Université de Sherbrooke, Sherbrooke QC J1E 4K8, Canada; seyedeh-raheleh.niavarani@usherbrooke.ca (S.R.N.); christine.lawson@usherbrooke.ca (C.L.); 2Centre de Recherche du Centre Hospitalier de l’Université de Sherbrooke, Sherbrooke QC J1E 4K8, Canada

**Keywords:** immunotherapy, oncolytic virus, autologous cancer vaccines, infected cell vaccines, natural killer cells, immunomonitoring

## Abstract

Oncolytic viruses (OVs) are a form of immunotherapy that release tumor antigens in the context of highly immunogenic viral signals following tumor-targeted infection and destruction. Emerging preclinical and clinical evidence suggests that this in situ vaccine effect is critical for successful viro-immunotherapy. In this review, we discuss the application of OV as an infected cell vaccine (ICV) as one method of enhancing the potency and breadth of anti-tumoral immunity. We focus on understanding and manipulating the critical role of natural killer (NK) cells and their interactions with other immune cells to promote a clinical outcome. With a synergistic tumor killing and immune activating mechanism, ICVs represent a valuable new addition to the cancer fighting toolbox with the potential to treat malignant disease.

## 1. Introduction

While the field of oncology has seen great advances in treating primary solid cancers, malignant cancers that have spread to multiple sites of the body have rarely been cured. Surgery, chemotherapy, and radiation are the mainstay of treatment for solid cancers, but the majority of patients bearing solid tumors develop metastasis after surgical resection and adjuvant treatment [1,2]. Depending on the solid cancer type, adjuvant chemotherapy and targeted therapies can prolong survival in metastatic cancers, but a cure is rarely achieved. According to the World Health Organization, cancer is a global healthcare concern, representing 14 million new cases each year and one of the deadliest diseases, with more than 8 million deaths annually, mainly attributable to metastatic cancer (WHO reports, 2018). Therefore, there is a pressing medical need for the development of new therapies to treat metastatic cancer. 

## 2. Checkpoint Inhibitors and the Emergence of Cancer Immunotherapies

The immune system plays a critical role in determining the therapeutic outcome of most solid cancers. Countless groups have demonstrated that patients who mount an immune response against their cancer as evidenced by the presence of tumor infiltrating lymphocytes (TILs) have a markedly improved prognosis in most solid cancers [3,4,5,6,7]. The successes seen in checkpoint inhibitor monoclonal antibodies has reignited enthusiasm for the field of cancer immunotherapy. Moreover, checkpoint inhibitor immunotherapy has revealed that solid cancers fall into two groups: “hot tumors”, which contain abundant anti-tumor T cells and respond for the most part to checkpoint inhibitor therapy; and “cold tumors”, which contain very little or no anti-tumor T cells and generally do not respond to checkpoint inhibitors [8]. Melanoma is the prototypic hot tumor. Melanoma cells generally contain abundant somatic mutations, which leads to neoantigen expression and recognition by the host’s immune system as foreign. This leads to the influx of TILs directed against those neoantigens. Mechanistic studies have revealed that in certain instances, T cells present in melanomas were unable to lyse tumor cells because of the expression of checkpoint molecules such as PD-L1 on the tumor cell surface to anergize the T cells [9]. The use of checkpoint inhibitor monoclonal antibodies interfered with this mechanism, allowing the anergized T cells to lyse tumour cells expressing cognate antigens. However, only 10%–50% of melanoma patients experience clinical response to checkpoint inhibitors and of those, 10%–15% of treated patients display adverse autoimmune syndromes [9].

On the other hand, the majority of solid cancers are considered “cold tumors” because they have very little or no pre-existing TILs [10]. Two possible reasons for this are the lack of immunogenicity of tumor associated antigens (TAAs) and immunosuppression in the tumor microenvironment (TME) [8]. TAAs by themselves do not trigger robust immune responses because they are either mutated self-antigens or overexpressed, non-mutated self-antigens against which the immune system has developed tolerance [11,12]. Furthermore, many tumors downregulate their expression of major histocompatibility complex (MHC) or costimulatory molecules, which are both needed to generate a robust adaptive immune response.

## 3. Oncolytic Viruses Can Make Cold Tumors Hot

Oncolytic viruses (OVs) are an emerging class of immunotherapeutic drugs. The origins of OVs arise from unusual observations of cancer remission in patients who contracted infectious diseases [13,14]. OVs were first approved for clinical use in head and neck cancers in China back in 2005, and more recently received FDA approval in the United States for late stage unresectable melanoma in 2015 [15]. The tumor-targeted ability of OVs resides in its exploitation of cancer mutations that facilitate oncogenesis, such as disrupted interferon signaling pathways [16]. Although OVs infect both normal and cancer cells, the intact interferon response of healthy cells fights off the OV infection. On the other hand, interferon pathway mutations in cancer cells allow for unchecked OV replication and oncolysis, leading to viral spread and cytotoxicity throughout the tumor [17,18,19].

The effectiveness of OV therapy depends not only on its ability to directly lyse cancer cells (oncolysis), but also on its initiation of anti-tumor immune responses. Most solid tumors exist as immune privileged sites, meaning they contain dynamic immune mechanisms that maintain immunosuppression. While this promotes tumor escape by evading the host immune system, it simultaneously allows OV infection with limited host immune interference. Therefore, OV-induced oncolysis leads to the release of TAAs in the context of highly immunogenic signals created by the virus infection. Thus, OVs can be more suitably described as oncolytic or in situ vaccines [17,18,20,21].

## 4. OV Infected Tumor Cells Reveal Neo-Epitopes through Immunogenic Cell Death (ICD), Leading to Anti-Tumor Immune Responses

In situ vaccination represents a strategy to exploit existing TAAs available at a tumor site without the need to previously identify and isolate them, which is a costly and time-consuming endeavour. Multiple studies have shown that OVs have the ability to induce viral-mediated immunogenic cell death (ICD) [14,17,22]. The infection of tumors by OVs leads to the release of pathogen associated molecular patterns (PAMPs) and danger associated molecular patterns (DAMPs), which signal through toll-like receptors (TLR) and activate cellular stress, which ultimately leads to the release of TAAs in a T cell inflamed TME capable of recruiting and activating antigen presenting cells (APCs), such as dendritic cells (DC) and critical anti-tumor effector cells such as natural killer (NK) and T cells [23].

Certain types of pre-mortem cellular stress and subsequent cell death trigger the release of specific molecules (PAMPs and DAMPs) that have been linked to the immunogenicity of cell death. Danger signaling includes the unfolded protein response resulting in the surface exposure of calreticulin (ecto-CRT) and other endoplasmic reticulum (ER) chaperones (HSP70 and HSP90), induction of autophagy and release of adenosine triphosphate (ATP), cell death and release of high mobility group box 1 (HMGB1), induction of the type I interferon (IFN) response, and production of CXC-chemokine ligand 10 (CXCL10/IP10) [22,24,25,26]. These DAMPs signal through endogenous pattern recognition receptors (PRRs), including TLRs, nod-like receptors (NLRs), and retinoic acid inducible gene (RIG-1)-like receptors (RLRs). Specific PRR signaling induces DC recruitment, maturation/activation, antigen uptake and processing, and signaling via the inflammasome with secretion of IL1β, crucial for priming and activation of tumor-specific T cells [25]. For example, ecto-CRT stimulates the engulfment of dying cancer cells by macrophages and DCs, which is critical for antigen cross-presentation; and extracellular HMGB1 acts on TLR4, leading to maturation and optimal tumor antigen processing by DCs [27]. The stress-inducible heat shock protein (HSP) 70 is know to function as an endogenous DAMP that can increase the immunogenicity of tumors and induce immune effector cell responses, including NK cells through NKG2D ligand cooperation [28].

While cancer cells can undergo ICD following radiation and chemotherapy, this represents a form of sterile inflammation, whereas infection with an OV provides infectious inflammation, which is likely more stimulatory to our immune system [29,30]. Indeed, all OVs activate features of ICD, and often by different mechanisms. OV infection of tumor cells releases both DAMPs and PAMPs, including virions, viral RNAs, or hypomethylated DNA, which signal through the same family of PRRs [23,31]. These strong signals of adjuvanticity are combined with virally-induced lysis of the infected cells, leading to the release of neoepitopes of cross-presentation by DCs. As well, direct infection of DCs by some OVs, including rhabdoviruses, further aides DC maturation and antigen presentation, ultimately leading to the activation and recruitment of additional innate and adaptive immune effectors [32,33].

## 5. OV-Based Infected Cell Vaccines (ICVs) Exploit the Properties of ICD Propagated through Viral Replication

Infected cell vaccines (ICVs) represent a potentially powerful personalized cancer therapy platform that exploits the efficacy of OVs and autologous cancer vaccines. Infusion of inactivated whole cancer cells, extracted from a patient’s own cancer (autologous), represents a truly personalized treatment option for malignancies with no targeted therapies [11]. Treatment with autologous cancer cells exposes a patient to their complete and individualized TAA repertoire. Therefore, the multiple epitopes contained within the vaccine can potentiate a polyclonal immune response capable of recognizing and eliminating a more diverse population of heterogeneous tumor cells [12]. However, whole cancer cell vaccines have demonstrated limited success in clinical trials, mainly owing to a lack of immunogenicity. The combination of cytokine delivery with tumor cells is capable of significantly delaying tumor growth through the creation of a pro-inflammatory environment to enhance immune system activation against TAAs. Existing data suggest that disease recurrence is significantly delayed when patients successfully mount an immune response against the tumor, as evidenced by a delayed-type hypersensitivity response [34]. Unfortunately, the majority of patients do not mount such a response, either because the cell vaccine and cytokine combination is not immunogenic enough or because the host immune system is suppressed in response to the cancer [35].

When cancer cells are infected by OVs in vivo, they effectively create an in situ ICV. The recently approved herpes based OV, T-Vec, is administered by intra-tumoral delivery and is effectively an in situ ICV [15]. An ex vivo ICV, however, consists of harvesting a patient’s tumor, followed by ex vivo infection with replicating OV and re-administration directly back into the tumor bed/tumor microenvironment. An ex vivo ICV approach is more cumbersome, but avoids some of the barriers to OV delivery (for example, neutralizing antibodies in the serum) and allows ex vivo modification of the cancer cells to improve adjuvanticity.

Triggering specific combinations of PRRs in APCs can induce a synergistic production of cytokines, leading to an initial NK cell response followed by a more robust antigen-specific CD8^+^ T cell mediated immune response. Rhabdoviruses are negative sense, single stranded RNA viruses that activate RLRs, TLR3, TLR7, and their G-protein signal through TLR4 [36]. HSV is a DNA virus that activates RLRs, TLRs, STING, and the inflammasome, while virion glycoproteins signal via TLR2 and entry activates intrinsic stress pathways [37,38]. Vaccinia virus signals predominantly through TLR2, and the RLRs RIG-1 and MDA5 [39]. Consistent with the ability to activate diverse DAMPs, PAMPs, and PRRs, different OVs induce different degrees of ICD. Similarly, there is strong precedent for combining OVs with enhancers of ICD or PRR signaling, including HSV combined with mitoxantrone or bortezomib, adenovirus combined with temozolomide and cyclophosphamide, and parvovirus and gemcitabine [13]. However, when combining OVs with other ICD inducers or PRR agonists, one must not only take into consideration the potential to skew the immune response toward immunosuppression, but must also avoid preferentially enhancing the immune response against the OV, potentially limiting therapeutic efficacy.

## 6. Evidence for NK Cell Activation with OVs and ICVs

Viruses, in general, activate NK cells [40,41] and OVs are no exception. One of the first studies to support the anti-tumor activation of NK cells in response to OV therapy was reported by Diaz et al., where depletion experiments were conducted to demonstrate that B16 melanoma tumor regression was achieved in a NK and CD8^+^ T cell dependent manner following vesicular stomatitis virus (VSV) intratumoral (i.t.) injection [18]. In support of these findings, oncolytic reovirus treatment of prostate cancer resulted in prominent NK cell infiltration and activation [42,43]. Miller et al. also observed that i.t. therapy with oncolytic herpes simplex virus (HSV) for B16 melanoma was abrogated in syngeneic murine models lacking NK and T cell subsets [44]. In mechanistic studies using oncolytic New Castle disease virus (NDV), Jarahian et al. demonstrated enhanced NK cell-mediated cytotoxicity against human tumor cell lines infected with NDV. Moreover, blocking assays demonstrated that NKp44 and NKp46 recognition of viral ligand hemagglutinin–neuraminidase on NDV infected tumor cells mediated NK cell antitumor activity [45]. We have demonstrated that the oncolytic ORF virus has a profound effect on NK cell cytotoxicity and cytokine secretion following i.v. delivery, and that this NK cell functionality is the main mechanism by which oncolytic parapoxvirus ovis (ORFV) exerts its anti-tumour effect [46].

We recently demonstrated a similar anti-tumor effect with the novel oncolytic rhabdovirus, maraba (MG1). MG1 is a double mutant rhabdovirus with deletion in the G and M proteins [19]. It is a clinical candidate OV that is currently in phase I/II clinical trials (NCT02285816, NCT02879760). We observed that MG1 infection in immune competent mice resulted in an immediate (24 h) and intense activation of NK cells, as evidenced by significantly increased NK cell cytotoxicity and cytokine secretion. Moreover, preoperative i.v. administration of MG1 overcame surgery induced NK suppression and attenuated the development of postoperative metastases in the B16lacZ model of implanted lung metastases, as well as in the breast 4T1 model of spontaneous lung metastases [47]. Mechanistically, we showed that MG1 activates NK cells through conventional DC (cDC). Using an *ex vivo* NK/DC co-culture system, we showed a lack of NK infection, activation, and cytotoxicity in the absence of cDC. Further, in cDC ablated mice, NK cell cytotoxicity was significantly reduced following MG1 administration [47]. While we demonstrated that MG1 does not directly infect or activate NK cells, this is not the case for other OVs. For instance, vaccinia virus has been shown to interact directly with NK cells through toll-like-receptor-(TLR)-2 [39]. It is very likely that stimulation of NK cells plays an important role in the therapeutic effect of many OVs, not only by enhancing NK cell mediated killing of tumour target cells, but also by triggering a robust, T cell-mediated, anti-tumour immune response [48].

Our laboratory and others have endeavored to improve upon the immunogenicity of the autologous cancer vaccination paradigm by infecting autologous cancer cells ex vivo with OVs and recombinant OVs engineered to express immune modulating cytokines [31,49,50,51].

Shirrmacher et al. provided the first preclinical evidence for this approach by infecting irradiated murine ESb tumor cells with oncolytic NDV. They demonstrated that vaccination with NDV-infected tumor cells was able to protect 50% of syngeneic mice from postoperative metastatic disease. These observations were further confirmed in B16 melanoma, 3LL Lewis Lung Carcinoma, and guinea pig L10 hepatocellular carcinoma models [52,53]. Notably, in clinical studies, 10-year follow-up results from a randomized-controlled phase II/III study in colon cancer patients with liver metastases performed by the same group showed significant advantages for vaccinated patients (receiving six injections of NDV infected autologous cancer cells) with respect to overall survival (*p* = 0.042) and disease-free survival (*p* = 0.047) over the control arm. In contrast, no treatment benefits were observed in rectal carcinoma patients on the same trial [54]. Although these clinical results are promising, future investigations with immune monitoring including NK cells are required to understand the efficacy of NDV-infected tumor cells as well as the biological differences between the two solid tumor types.

Using oncolytic rhabdovirus vesicular stomatitis virus harboring a deletion in the M protein (VSV-Δ51) in the B16 melanoma model, Lemay et al. demonstrated that a prime and boost immunization strategy, seven days apart, with the VSV-Δ51 infected B16 tumor cells was able to completely protect 30% of the C57Bl/6 mice from a B16 subcutaneous tumor challenge. Moreover, when a VSV-Δ51 expressing granulocyte macrophage-colony stimulating factor (GM-CSF) was used for the ICV, potent activation of both NK cells and T cells was observed in addition to tumor debulking and long-term cancer surveillance [49]. Conrad et al. demonstrated similar efficacy and immunity using an *ex vivo* ICV made with the closely related rhabdovirus Maraba MG1 in an aggressive L1210 murine leukemia model [55].

We recently demonstrated that the intratumoral delivery of autologous colon cancer cells infected *ex vivo* with maraba MG1 containing an IL12 transgene (MG1-IL12-ICV) provided a significant therapeutic benefit to normally resistant mouse models of established peritoneal disease [50]. MG1-IL12-ICV was well tolerated by mice while inducing a robust recruitment of cytotoxic NK and T cells to the peritoneal cavity [50]. Importantly, the highest treatment efficacy was observed in mice treated with MG1-IL12-ICV and not with parental MG1-ICV, or uninfected tumor cells, or MG1-IL12 virus used as an oncolytic agent alone. Even in mice with bulky peritoneal carcinomatosis (abdominal malignancies), a complete radiologic response was demonstrated within 8–14 weeks and was associated with 100% long-term survival.

## 7. The Importance of NK Cell Monitoring in OV and ICV Therapies

From preclinical OV and ICV studies, it is clear that NK cells play a key mediating role in the generation of antitumor immunity. This supports the idea that targeting both innate and adaptive immune mechanisms may synergistically promote a clinical outcome. The proposed mechanism of action of ICV relies on the capacity of recruited and activated DCs to present captured TAAs to T cells, which is essential for generating specific T cell immunity [56]. Oncolytic NDV was demonstrated to induce tumoricidal activity in NK cells by binding to NKp46 receptors and initiating activation signals leading to cytotoxic activity and IFN-γ production [45,54]. In our research, we have evaluated the immunological outcome of *ex vivo* ICV strategies based on induction of both NK and specific T cell responses. Our immune cell depletion studies have shown reduced survival in oncolytic rhabdovirus ICV treated mice depleted of NK cells [50]. Given the importance of NK/DC crosstalk in the development of an immune response, specific monitoring of NK cells and their responses should be pursued in ICV protocols. As we have observed in our previous studies, oncolytic rhabdovirus ICV can stimulate the recruitment, activation, and cytotoxic activity of NK cells by soluble signals (for example, IP10), contributing to the development of potent innate immunity [47,50]. In turn, we hypothesize that these activated NK cells can further stimulate host DCs, contributing to the recruitment and activation of DCs to advance sustained antitumor T cell immunity (Figure 1).

The monitoring of NK cells and their contribution to the overall antitumor immune response is seldom included in human cancer immunotherapy trials. In a phase IIb clinical trial with oncolytic vaccinia virus JX549, we reported on the postoperative activation of NK cells at early time points following virus administration in colorectal cancer patients with liver metastases [57]. However, long-term follow up was not conducted in this trial to correlate early NK cell functionality and therapy outcome. To the best of our knowledge, NK cell monitoring has not been conducted in any existing ex vivo ICV human trials. While DC-based cancer vaccines are not the same as ICV, the central mechanism of action of both therapies is dependent on tumor antigen presentation by DC. Emerging clinical trials of DC-based cancer vaccines have revealed that activated NK cells are better predictive of vaccine efficacy than cytotoxic T lymphocyte responses [33,58,59,60,61]. In these DC-based cancer vaccines trials, NK cell immune monitoring includes assessment of peripheral blood NK cell proportion, total numbers, and phenotypic analysis before and after therapy [59,62]. Moreover, tumor infiltrating and lymphoid tissue NK cells have been evaluated [59]. NK cell functional assays usually focus on NK cell-mediated cytolytic activity against chromium labelled NK-sensitive cell lines including K562 and Daudi. NK cell-secreted cytokines [for example, IFN-γ, tumor necrosis factor-α (TNF-α), and interleukin-15 (IL-15)] have also heavily measured for DC-vaccine therapy induced effects. Notably, a few DC vaccine studies have performed specific lysis assays against autologous primary tumor cells to evaluate therapy efficacy [60,63], which is likely to be of high relevance in the setting of eliminating residual tumor cells to prevent relapse. Lastly, indirect NK cell-related parameters are the subject of recent research because a range of transformed cancer cells has been shown to downregulate or secrete soluble molecules relevant to NK cell function (for example, soluble ligands for the activating receptor NKG2D) in order to evade NK cell immunosurveillance [64,65]. In this regard, evaluation of soluble NKG2D ligands in the plasma of patients prior to and following DC-based vaccine therapy could be a significant predictive marker for therapy effectiveness.

## 8. Clinical Development Potential of OVs and ICVs

Across the field of oncology, immunotherapeutic agents such as OVs and therapeutic vaccines are poised to transform cancer treatment. There are currently two approved OV therapies targeting cancer—T-Vec since 2016 (AMGEN) and Onyx-015 since 2005 (ONYX Pharmaceuticals). T-Vec is a genetically engineered herpes virus that was approved in the United States and Europe for unresectable stage III and IV melanoma [15], while Onyx-015 is a genetically modified adenovirus that was approved for head and neck cancer in China in 2005 [14]. Currently, pharmaceutical companies have demonstrated a renewed interest toward OVs owing to the recent strides made in checkpoint immunotherapies. Specifically, Merck invested $394 million to acquire Viralytics for the development of oncolytic Coxackie virus, while $900 million was engaged by Bristol-Myers Squibb to develop an oncolytic adenovirus with PsiOxus Therapeutics [66]. According to Global Cancer Vaccine Market & Clinical Trial Insight 2025, it is estimated that the therapeutic cancer vaccine market will reach more than 15 billion by 2025 with a compound annual growth rate (CAGR) of 27% from 2015 to 2022 [67]. Importantly, progress in the cancer vaccination field resides in developing novel and more potent vaccine adjuvants (such as OVs) that generate effector immune responses capable of overcoming systemic and local suppressive mechanisms.

Medicines in Development: Immuno-Oncology, in 2017, listed 14 oncolytic virus therapies in phase I to III clinical trials sponsored by 14 different companies [68]. Of these, one phase I/II trial targets solid tumors, including esophageal/gastrointestinal and breast cancers. This ongoing study conducted by the Canadian Cancer Trials Group is assessing the safety, pharmacokinetic, and immune profiles in response to an oncolytic adenovirus followed by rhabdovirus-based vaccine bearing MAGE-A3 using a heterologous prime-boost strategy (NCT02285816). This is an in situ vaccine targeting a single TAA. The same prime boost strategy is being tested by the same group in combination with checkpoint inhibitors for non-small cell lung carcinoma (NSCLC) and HPV-associated cancers (NCT02879760). Clinicaltrial.gov lists 10 active phase I/II cancer vaccine trials targeting metastatic cancer. Among them are OncoVAX for colorectal cancer; ProstVAC for prostate cancer; and Vigil for ovarian cancer, breast cancer, sarcoma, non-small cell lung carcinoma (NSCLC), and melanoma.

## 9. Future Perspectives

The ICV strategy is a highly potent NK cell-stimulating platform. Unlike the growing body of evidence acquired through experimental research, there is a paucity of data reporting NK cell function in cancer vaccine trials and immunotherapy trials in general [59,60]. We have provided an overview of preclinical and translational studies that measured the influence of NK cell participation in antitumor immunity following OV- and ICV-based immunotherapeutic strategies. In summary, these studies highlight a mediating role for the cytotoxic and cytokine-secreting functions of NK cells in the development of ICV-mediated adaptive antitumor immunity. This calls for the implementation of NK cell monitoring in cancer immunotherapy, in particular for OV- and ICV-based therapies. Thoughtful choosing of defined NK cell parameters and protocols is required. Ultimately, this will contribute to a more complete understanding of therapy efficacy and correlation with outcome. NK cell immune monitoring has the potential to generate valuable information that could be exploited in the development of evidence-based and novel immunotherapies to improve prognosis for cancer patients.

## Figures and Tables

**Figure 1 viruses-11-00434-f001:**
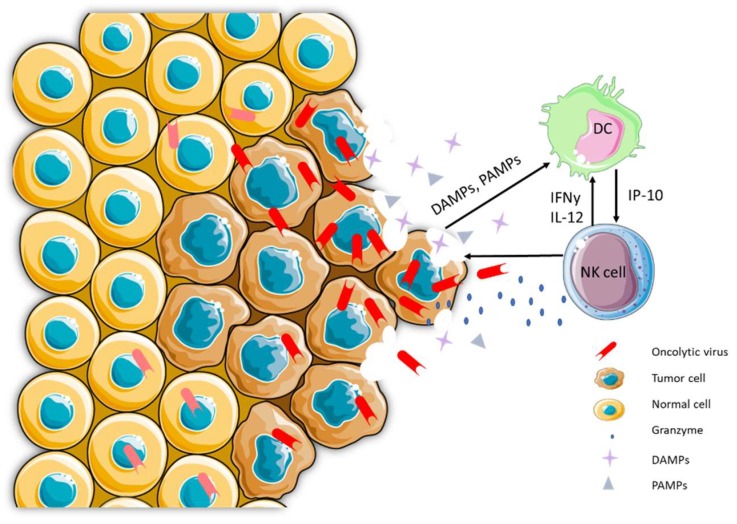
Oncolytic virus (OV)-infected tumor cells reveal tumor associated antigens through immunogenic cell death leading to anti-tumor immune responses. OVs take advantage of impaired antiviral and other essential signaling pathways in tumor cells, resulting in its selective growth advantage in tumor cells, but not in healthy cells. Infection and replication in tumor cells by OVs result in tumor cell lysis and the release of tumor associated antigens in the presence of highly immunogenic pathogen associated molecular pattern (PAMP) and danger associated molecular pattern (DAMP) signatures. This leads to the recruitment of innate and adaptive immune cells and the initiation of anti-tumor immune responses.

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
