# Peer review of "Treatment of Metastatic Disease through Natural Killer Cell Modulation by Infected Cell Vaccines"

_viruses, 2019, doi:10.3390/v11050434_

Round 1

Reviewer 1 Report

Oncolytic viruses are gaining traction as cancer immune-therapeutics due to their potential to induce immune responses against the tumor as a secondary effect of viral oncolysis. This review discusses the application of OVs as infected cell vaccines, with an emphasis on NK modulation as a crucial aspect of the vaccine approach. In general, I like the topic, and it is a novel deviation from the typical OV review. The manuscript is clearly written and easy to read (just a few grammatical errors), but I do have some (rather major) comments regarding the content which would need to be addressed:

1.      The authors introduce 2 categories of the infected cell vaccine (Section 5): the in situICV, where cancer cells are infected in vivo, and the ex vivoICV approach, in which tumor tissue is harvested, infected, and then infused back into the patient. None of the subsequent references to ICV specifies which approach they are referring to. Can I assume that they mean the ex vivoapproach, because the in situ vaccination is essentially just a standard OV therapy?  

2.     For a review on the topic of ICV, there is generally a lack of information about which viruses have been used as ICVs, how the vaccines were implemented, and what their respective efficacies are. Which viruses/protocols are most promising as ICVs?

3.     The title seems to overreach the data presented in the manuscript. Although the authors do discuss the importance of NK cells in the ICV approach, I could not find any evidence that NK cell modulation by ICVs prevents metastatic disease.  

4.     Section 7, describing the importance of NK cell monitoring in OV and ICV therapies, is a bit too vague. Are NK cells modulated in all OV/ICV approaches? I would guess that some viruses elicit a stronger NK cell response than others. The authors should specify which viruses were used in the studies they are referencing. With regards to the importance of NK cell monitoring, is there any data showing that the amplitude of NK cell #/function correlates with the outcome of the therapy? And for which viruses has this been investigated? Although I agree that the presence of NK cells is likely to be of great importance to the outcome, they are surely not the only players, and the authors have not provided enough information to explain why NK cell monitoring would be a better predictor of therapeutic outcome than, say, T cells or DCs. 

Author Response

We thank the reviewer 1 for their insightful comments, which has greatly improved the quality of our review manuscript.  Responses to the reviewer are in bold underneath each comment.

1.  The authors introduce 2 categories of the infected cell vaccine (Section 5): the in situ ICV, where cancer cells are infected in vivo, and the ex vivo ICV approach, in which tumor tissue is harvested, infected, and then infused back into the patient. None of the subsequent references to ICV specifies which approach they are referring to. Can I assume that they mean the ex vivo approach, because the in situ vaccination is essentially just a standard OV therapy? 

Yes, reviewer 1 is right.  The in situ vaccination approach is indeed standard OV therapy, while we are referring to the ex vivo approach in section 6 and onwards.  Where appropriate, we add the text “ex vivo” in front of descriptions of ICV.

2.  For a review on the topic of ICV, there is generally a lack of information about which viruses have been used as ICVs, how the vaccines were implemented, and what their respective efficacies are. Which viruses/protocols are most promising as ICV?

- An expanded section 6 detailing the oncolytic viruses that have been used for the ex vivo ICV approach, vaccine implementation and their respective efficacies has been added.

3.  The title seems to overreach the data presented in the manuscript. Although the authors do discuss the importance of NK cells in the ICV approach, I could not find any evidence that NK cell modulation by ICVs prevents metastatic disease

We have modified the title to “Treatment of metastatic disease through natural killer cell modulation by infected cell vaccines” to better represent the data presented in the review manuscript.

4.  Section 7, describing the importance of NK cell monitoring in OV and ICV therapies, is a bit too vague. Are NK cells modulated in all OV/ICV approaches? I would guess that some viruses elicit a stronger NK cell response than others. The authors should specify which viruses were used in the studies they are referencing. With regards to the importance of NK cell monitoring, is there any data showing that the amplitude of NK cell #/function correlates with the outcome of the therapy? And for which viruses has this been investigated? Although I agree that the presence of NK cells is likely to be of great importance to the outcome, they are surely not the only players, and the authors have not provided enough information to explain why NK cell monitoring would be a better predictor of therapeutic outcome than, say, T cells or DCs.

-  We have added more details in section 7 studies, including which viruses were used and their efficacies.  NK cells are rarely monitored in immunotherapy studies, even less so in OV trials, and not at all in the few ICV trials.  We, therefore, describe NK cell monitoring in dendritic cell-based vaccine approaches as a proxy for OV/ICV based approaches because their central mechanism of action (tumor antigen presentation) is similar.  Recent clinical trials of DC-based cancer vaccines have demonstrated that NK cells are better predictive of vaccine efficacy than CTL responses (references 57-60).

Reviewer 2 Report

The review article by Seyedeh Raheleh Niavarani et al described the application of OV as an infected cell vaccine (ICV) as one method of enhancing the potency and breadth of anti-tumoral immunity. This manuscript is well written and would be of interest to readers of Viruses. I recommend publishing this article.I recommend publishing this review article.

Author Response

We thank reviewer 2.

Round 2

Reviewer 1 Report

The revised version is acceptable for publication.